# Functional Similarity and Difference among *Bra-MIR319* Family in Plant Development

**DOI:** 10.3390/genes10120952

**Published:** 2019-11-21

**Authors:** Ziwei Hu, Tingting Liu, Jiashu Cao

**Affiliations:** 1Laboratory of Cell and Molecular Biology, Institute of Vegetable Science, Zhejiang University, Hangzhou 310058, China; 11416051@zju.edu.cn (Z.H.); 11416009@zju.edu.cn (T.L.); 2Key Laboratory of Horticultural Plant Growth, Development and Quality Improvement, Ministry of Agriculture, Hangzhou 310058, China; 3Zhejiang Provincial Key Laboratory of Horticultural Plant Integrative Biology, Hangzhou 310058, China

**Keywords:** *Brassica campestris*, Bra-miR319 family, leaf and petal morphogenesis, pollen development

## Abstract

miR319 was the first plant miRNA discovered via forward genetic mutation screening. In this study, we found that miR319 family members had similar sequences but different expression patterns in *Brassica campestris* and *Arabidopsis thaliana.* RT-PCR analysis revealed that *Bra-MIR319a* and *Bra-MIR319c* had similar expression patterns and were widely expressed in plant development, whereas *Bra-MIR319b* could only be detected in stems. The overexpression of each *Bra-MIR319* family member in *Arabidopsis* could inhibit cell division and function in leaf and petal morphogenesis. Bra-miR319a formed a new regulatory relationship after whole genome triplication, and *Bra-MIR319a* overexpressing in *Arabidopsis* led to the degradation of pollen content and affected the formation of intine, thereby causing pollen abortion. Our results suggest that *Bra-MIR319* family members have functional similarity and difference in plant development.

## 1. Introduction

MicroRNAs (miRNAs) are a class of short (about 21 nt in length) noncoding RNAs [1]. In plants, different miRNA precursor genes can form the same or similar mature sequences, and these precursor genes are considered to be the same family [2,3]. miRNA family members have similar functions due to the same or similar mature sequences. However, each precursor gene in the miRNA family has different expression patterns and targets, so they exhibit functional differentiation [4].

miR319 was the first plant miRNA discovered via forward genetic mutation screening; leaves of the mutant *jaw-D* are crinkly and serrated [5]. In *Arabidopsis thaliana*, three precursor genes of miR319 can produce two mature sequences, which differ by one base at the 3′ end [6]. The three members of *MIR319* have different expression patterns. *MIR319a* is continuously expressed during plant growth, *MIR319b* is expressed only during vegetative growth, and *MIR319c* is highly expressed in reproductive growth [7]. Different expression patterns lead to differences in gene function. The overexpression of *MIR319a* results in leaf phenotypes similar to those of *jaw-D* mutants, whereas leaves of overexpressing *MIR319c* plants have no difference with those of wild-type plants [6]. The expression level of miR319a is higher than those of miR319b and miR319c. Thus, the current report on the regulation of the miR319 family mainly focuses on miR319a. miR319a can target five TCP transcription factors but also some MYB transcription factors in *A. thaliana* [6,7]. miR319a participates in leaf morphogenesis by targeting *TCPs* [8,9]. Given the cross-target of miR319 and miR159, the overexpression of *MIR319a* causes anther defects similar to those seen in overexpressing *MIR159a* plants [6]. High-throughput sequencing has revealed that miR319 is also present in pollen [10], but its role in pollen development is unclear.

*Brassica campestris* (syn. *B. rapa*) and *A. thaliana* are closely related in taxonomy [11]. Compared with *A. thaliana*, *B. campestris* undergoes whole-genome triplication (WGT), its genome is divided into three subgenomes, and the number and expressions of genes are differentiated in three subgenomes [12]. Similar to the encoded protein gene family, miRNA families also undergo functional retention and differentiation in *B. campestris* after WGT [13,14]. Moreover, the complexity of miRNAs regulating target genes is increased after gene duplication. miRNA target genes undergo a high degree of differentiation, and some miRNA binding sites are lost [15,16].

We previously used the precursor gene of miR319 in *A. thaliana* to perform BLAST in the Brassica database (http://brassicadb.org/brad/) and obtained the precursor genes of Bra-miR319 [17]. The similarity of miR319 precursor sequences between *B. campestris* and *A. thaliana* was over 80%. Bra-miR319a and Bra-miR319b had the same sequence, whereas Bra-miR319c had one base difference at the 3′ end. They shared the same sequences with those of *A. thaliana*. In this study, we found that although their sequences were highly conserved in *B. campestris* and *A. thaliana*, Bra-miR319 family members had different expression patterns than those in *A. thaliana*, and they could all function in leaf and petal morphogenesis. Moreover, Bra-miR319a could also target *BcMYB101* and function in pollen development. Our results indicate that after WGT, *Bra-MIR319* family members had functional similarity and difference in plant development.

## 2. Materials and Methods 

### 2.1. Plant Materials

Chinese cabbage “Aijiaohuang” (*B. campestris* L. subsp. *chinensis* Makino cv. Aijiaohuang) was cultivated in the experimental farm of Zhejiang University. *A. thaliana* (Col-0), transgenic plants, and nuclear localization tobacco (*Nicotiana benthamiana*) were grown under long-day conditions (16 h light/8 h dark) at 22 °C.

### 2.2. Expression Profile Analysis

The total RNA and small RNA from roots, stems, leaves, inflorescences, and siliques were extracted by using a microRNA extraction kit (Invitrogen, Carlsbad, CA, USA) and reversed transcribed into the first strand of cDNA through the PrimerScript RT reagent kit (TAKARA, Shiga, Japan). *BcUBC10* and *BcU6* were selected as the reference genes for RT-PCR and qRT-PCR; the gene-specific primers are listed in Appendix A. qRT-PCR was performed by using the SYBR^®^ Premix Ex Taq™ Kit (TaKaRa, Shiga, Japan) in a CFX96 Real-Time System (Bio-Rad, Hercules, CA, USA). The 2^−△△Ct^ method was used to compute the expression levels of different genes [18].

### 2.3. Generation of Bra-MIR319-Overexpression Arabidopsis Lines

The 168 bp precursor gene of *Bra-MIR319a*, 169 bp precursor gene of *Bra-MIR319b*, and 161 bp precursor gene of *Bra-MIR319c* were amplified with gene-specific primer pairs (Appendix A) and subcloned into the pBI121 vector with the constitutive *CaMV 35S* promoter, respectively. These constructs were transferred into the *Arabidopsis* plant through the floral-dip method mediated by *Agrobacterium* [19]. The seeds of transformed *Arabidopsis* were screened in 1/2 MS agar plate containing kanamycin. The total RNA from the transgenic plants and wild-type plants were extracted and the expression levels of *Bra-MIR319a*, *Bra-MIR319b*, and *Bra-MIR319c* were detected via qRT-PCR in accordance with the abovementioned method. Transgenic T_1_ plants were used for the phenotypic observation of vegetative growth and T_2_ plants were further used for the observation of pollen development.

### 2.4. Morphological and Cytological Observation

The methods for scanning electron microscopy (SEM) and transmission electron microscopy (TEM) observation were previously described [20]. Semithin section observations of leaf and anther were previously described [21]. Image-Pro software was used to measure the size of epidermal cells at 1 K magnification. Alexander staining was used to observe pollen vitality.

### 2.5. Cleavage Site of miRNA Targets

The miRNA target gene prediction website (http://plantgrn.noble.org/psRNATarget/) was used to predict the target genes of Bra-miR319. Total RNA was purified to extract poly (A) RNA. cDNA was synthesized using the RACE kit (TAKARA). Nested PCR was performed for 5′ rapid amplification of cDNA ends (RACE) with specific reverse primers (Appendix A).

### 2.6. Transient Expression in N. benthamiana

The CDS of *BcMYB101* was cloned into pFGC-eGFP to create the fusion construct by using gene-specific primer pairs (Appendix A). As a control, we amplified the CDS of *mBcMYB101* with multiple mutations in the miRNA complement motif, preventing the miRNA from cleaving without altering its amino acid sequence, and cloned it into pFGC-eGFP via overlapping PCR. The method of coexpression of Bra-miR319a and *BcMYB101* was described previously [6]. After 48 h of transfection, GFP fluorescence was detected using a confocal laser scanning microscope (Nikon A1-SHS, Japan). 

## 3. Results

### 3.1. Bra-MIR319 Family Members Have Functional Similarity in Leaf and Petal Morphogenesis

Expression of Bra-miR319 was detected via qRT-PCR in different tissues of *B. campestris*, including roots, stems, leaves, inflorescences, and siliques. The result showed that Bra-miR319 was expressed in all tissues, and its expression in the stem was higher (Figure 1a). Given that Bra-miR319 could be produced by different precursors, the expression of these precursors was detected via RT-PCR. The expressions of *Bra-MIR319a* and *Bra-MIR319c* were similar, and *Bra-MIR319b* was only expressed in stems (Figure 1b). We constructed overexpression vectors for *Bra-MIR319a*, *Bra-MIR319b*, and *Bra-MIR319c*, respectively, and transferred them into the *Arabidopsis* plants to observe the phenotype of transgenic plants. Expressions of *Bra-MIR319a*, *Bra-MIR319b*, and *Bra-MIR319c* were all increased in transgenic *Arabidopsis* plants (Appendix A). p35S: *Bra-MIR319a*, p35S: *Bra-MIR319b*, and p35S: *Bra-MIR319c* transgenic *Arabidopsis* plants could all show crinkly leaves (Figure 2b–d). Further, transgenic *Arabidopsis* plants also showed wavy and yellow-green petals (Figure 2e–h). Compared with the wild-type petal, the petal vascular tissue of the transgenic plants exhibited an irregular shape (Figure 2i–l). These results indicated that all precursor genes of Bra-miR319 could be efficiently processed in *Arabidopsis*, and they had similar functions in leaf and petal morphogenesis.

### 3.2. Bra-MIR319 Family Members Overexpressing in Arabidopsis Can Inhibit Cell Division 

We selected the fifth leaf of the fully expanded transgenic and wild-type *Arabidopsis* plants for further cytological observation. SEM showed that the overexpression of *Bra-MIR319a*, *Bra-MIR319b*, and *Bra-MIR319c* reduced the number of epidermal cells in *Arabidopsis* leaves, but the cells increased in size (Figure 3a–d,i). The size of the epidermal cells of wild-type and transgenic plants were measured by Image-Pro software at 1 K magnification. The size of the epidermal cells of the wild-type plants was 1258.8 ± 545.8 μm^2^, and the size of the epidermal cells of *Bra-MIR319* family members overexpressing plants was 1835.1 ± 666.1, 2612.5 ± 152.6, and 1687.2 ± 518.2 μm^2^, respectively. Comparing the shape of leaf epidermal cells, it was found that cells of the wild-type plants were "jigsaw", while the epidermal cells of the transgenic plants were rounder and showed undifferentiated cell morphology. The results of semithin section analysis showed that the number and arrangement of palisade tissues in leaves of transgenic plants were not different from those of wild-type plants, but the number of sponge tissues was significantly less than that of wild-type plants (Figure 3e–h,j–k). These results indicated *Bra-MIR319a*, *Bra-MIR319b*, and *Bra-MIR319c* were involved in cell division and differentiation.

### 3.3. Bra-MIR319a Overexpressing in Arabidopsis Reduces Pollen Viability

We further observed the reproductive growth of transgenic plants. We examined the pollen viability and morphology of *Bra-MIR319a*, *Bra-MIR319b*, and *Bra-MIR319c* overexpressing plants. Alexander staining results showed that nearly 86.4% of the pollen in the p*35S:Bra-MIR319a* anthers was blue-green, and only a small part of the pollen was viable. In contrast, the pollen stainings of *Bra-MIR319b* and *Bra-MIR319c* transgenic plants were indistinguishable from those of wild-type plants, and both pollen grains were viable (Figure 4a–d). The pollen morphology was observed by SEM and the results showed that the pollen morphology of *Bra-MIR319b* and *Bra-MIR319c* transgenic plants did not differ from that of wild-type plants (Figure 4g–h and k–l). However, these aborted pollen grains of *Bra-MIR319a* overexpressing plants were shrunk (Figure 4f,j). Thus, the results showed that *Bra-MIR319* family members had functional differentiation in pollen development, and the overexpression of *Bra-MIR319a* could lead to remarkable pollen abortion.

### 3.4. Bra-MIR319a Overexpressing in Arabidopsis Affects the Formation of Pollen Content and Intine

We observed anthers at different developmental stages via semithin section analysis to clarify the underlying reasons for pollen abortion in *Bra-MIR319a* overexpressing transgenic *Arabidopsis* plants. Until the early uninucleate stage, no difference was observed between *Bra-MIR319a* transgenic plants and wild-type plants (Figure 5a–b,e–f). The results of semithin sectioning revealed that at the binuclear stage, *Bra-MIR319a* transgenic plants showed vacuolar pollen, and the vacuolar pollen completely shrunk at the mature pollen stage (Figure 5g–h). TEM was used for more detailed observation. The results of TEM showed that the pollen content of *Bra-MIR319a* transgenic plants began to degrade from the uninucleate vacuole stage and continued to degrade in the binuclear stage, with complete degradation during the mature pollen stage (Figure 6f–h). With the degradation of the pollen content, the pollen intine of the *Bra-MIR319a* transgenic plant disappeared with the intact structure of pollen exine (Figure 6i–j). These findings suggest the overexpression of *Bra-MIR319a* led to the degradation of pollen contents, thereby affecting the formation of pollen intine and leading to pollen distortion.

### 3.5. Bra-miR319a Can Target Homologs of MYB101

In *Arabidopsis*, since miR319a also targeted *MYB33* and *MYB65*, some overexpression of miR319a plants could lead to additional stamen defects resembling those of *35S:miR159a* plants. However, after WGT, homologs of *MYB33* were not detected in any of the *Brassica* species and the miRNA-binding region was entirely missing in homologs of *MYB65* in *B. campestris* [6,22]. According to the analysis results of the miRNA target gene prediction website, we found Bra-miR319a could target homologs of *MYB101*. In *B. campestris**,* homologs of *MYB101* had three copies: *BcMYB101-1* (Bra022888), *BcMYB101-2* (Bra021791), and *BcMYB101-3* (Bra005597). The results of the 5′ RACE experiment proved that Bra-miR319a could cleave *BcMYB101-1* between nucleotides 10 and 11 of the miRNA recognition sequence (Figure 7a). Moreover, we transiently expressed the BcMYB101-1: GFP fusion protein in *N. benthamiana* leaves and coexpressed it with Bra-miR319a. As expected, the coexpression of Bra-miR319a led to the disappearance of GFP fluorescence (Figure 7c). As a control, we prepared a version of BcMYB101-1: GFP with multiple mutations in the miRNA complementary motif (Figure 7b), and when coexpressed with Bra-miR319a, the GFP signal still existed (Figure 7c). These data indicated that the regulation of target genes by Bra-miR319a was differentiated after WGT.

## 4. Discussion

In Brassicaceae, *MIR319a* and *MIR319b* were closely related and formed sister clades, with *MIR319c* forming a separate clade [23]. In *Arabidopsis*, the expression of *MIR319a* was high during plant development. *MIR319a* overexpressing plants had crinkly leaves, whereas those with *MIR319c* overexpression had normal leaves due to limited expression during vegetative growth. Therefore, miR319a played a major role in leaf development [6,7]. In contrast to the expression pattern of *MIR319* members in *Arabidopsis*, *Bra-MIR319a* and *Bra-MIR319c* had similar expression patterns, whereas *Bra-MIR319b* was only expressed in the stem (Figure 1b). The overexpression of *Bra-MIR319c* could produce a leaf phenotype similar to that of plants overexpressing *Bra-MIR319a* (Figure 2a). The overexpression of *Brassica oleracea MIR319c* in *Arabidopsis* could also produce crinkled leaves, whereas the overexpression of *B. oleracea MIR319a* resulted in normal rosette leaves because the mature sequence of *Bol-MIR319a* differed in a single nucleotide from others. Thus, the function of the *MIR319a* gene in *B. oleracea* was assumed by *Bol-MIR319c* or *Bol-MIR319b* [24]. However, the mature sequence of *MIR319a* in *B. campestris* was the same as that of *A. thaliana* and could lead to similar crinkled leaves. Thus, *Bra-MIR319* family members had functional similarity in leaf development and could all give rise to functional Bra-miR319 in *A. thaliana*.

In *Arabidopsis*, miR319a could target some TCP transcription factors to inhibit cell division through different pathways [8,9]. In our study, SEM showed that the number of epidermal cells in *Bra-MIR319* family member transgenic plant leaves was reduced, and the shapes of epidermal cells were undifferentiated (Figure 3b–d). According to the miRNA target gene prediction website, Bra-miR319 family members could also target homologs of *TCP3* and *TCP4*, so we presumed that Bra-miR319 had similar regulatory pathways as those in *Arabidopsis* to affect cell division and differentiation. In addition, miR319-regulated *TCP4* could directly activate *YUCCA5* and integrate the auxin response to promotes cell elongation in *A. thaliana* hypocotyls [25]. Overexpression of *Bra-MIR319* family members could also cause the size of the epidermal cells to increase, but it needed more experiments to illustrate the function of the *Bra-MIR319* family in the auxin pathway. Bra-miR319 family members had more functions in petal morphogenesis. Petals could be compared with the leaf lamina because both were broadly extended and positioned distally [26]. Thus, the overexpression of Bra-miR319 family members could also cause wavy petals and even cause the petals to darken (Figure 2i–l). In *Arabidopsis*, overexpression of *MIR319a* could affect the accumulation of chlorophyll and delay leaf senescence, and in double mutant *mir319ab*, the content of chlorophyll began to decline earlier than the wild-type plants [7,27]. So, we speculated the dark green color of the petals of *Bra-MIR319* overexpressing plants might be caused by chlorophyll accumulation.

*Bra-MIR319a* could be expressed in inflorescence and caused 86% pollen abortion. *Bra-MIR319a* overexpressing in *Arabidopsis* gave rise to the degradation of pollen contents from the uninucleate stage, and the pollen intine of transgenic plants disappeared (Figure 6f–j). Pollen contents provided the raw material for the formation of pollen intine [28]. During the uninucleate stage, pollen intine started forming and thickened in the binuclear stage. Thus, the pollen grain remained normal in form; however, abnormal development of pollen intine caused the pollen grain to shrink and led to abortion [29]. In *Arabidopsis*, except *TCPs*, miR319 could also target *MYB33* and *MYB65* and caused anther defects similar to miR159 [6]. However, after WGT in *B. campestris*, homologs of *MYB33* were lost [6,21]. In this study, we confirmed that Bra-miR319a could target homologs of *MYB101* (Figure 7). *MYB101*, *MYB120*, and *MYB97* had functional redundancy in pollen tube growth in *Arabidopsis* [30]. In *B. campestris*, homologs of *MYB97* were lost, and *BcMYB101* and *BcMYB120* each had three copies, whereas Bra-miR319a could only target one copy of *BcMYB101*. Thus, we inferred that they might exhibit functional differentiation. Bra-miR319a might function in pollen development by regulating *BcMYB101-1.* In our future work, we will investigate the relevant detailed mechanisms. Although *Bra-MIR319c* had similar expression in the inflorescence, it did not cause pollen abortion when overexpressed in *Arabidopsis*. However, when *Bra-MIR319c* was overexpressed in *B. campestris* ssp. *chinensis* cv. Youqin 49, it could cause 25% pollen abortion [17]. We suspect that Bra-miR319c played a minor role in pollen development.

In conclusion, *Bra-MIR319* family members had different expression patterns, but they had functional similarity in leaf and petal morphogenesis. Moreover, we revealed that the overexpression of *Bra-MIR319a* in *Arabidopsis* played a role in pollen intine formation and caused pollen grains to shrink. This study helps us to deepen our understanding of the similarity and difference of miRNA function in the same family and to explore new miRNA regulatory networks formed after replication events.

## Figures and Tables

**Figure 1 genes-10-00952-f001:**
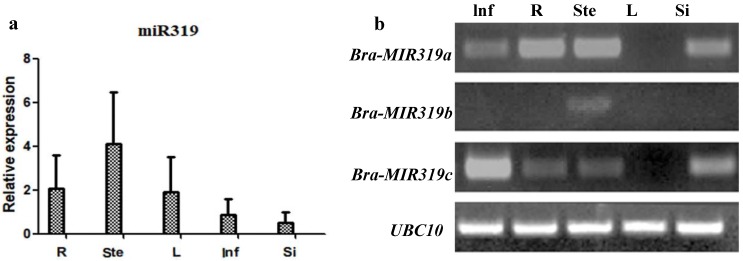
Expression pattern of Bra-miR319 family members. (**a**) Quantitative real-time PCR analysis of Bra-miR319 in different tissues of *Brassica campestris*. (**b**) Real-time PCR analysis of precursor genes of Bra-miR319 in different tissues of *B. campestris*. Roots (R), stems (Ste), leaves (L), inflorescences (Inf), and siliques (Si).

**Figure 2 genes-10-00952-f002:**
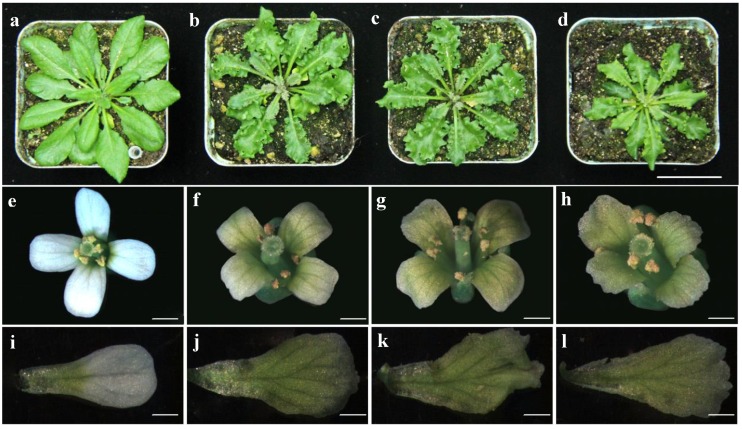
Phenotype of *Bra-MIR319* family overexpressing transgenic *Arabidopsis*. (**a**–**d**) Rosette plants of wild-type and overexpressing the *Bra-MIR319a*, *Bra-MIR319b*, and *Bra-MIR319c*, respectively. (**e**–**h**) The flower of wild-type plant and transgenic *Arabidopsis* overexpressing the *Bra-MIR319* family, respectively. (**i**–**l**) The petal of wild-type plant and transgenic *Arabidopsis* overexpressing the *Bra-MIR319* family, respectively. Scale bars, 2 mm in (**a**–**d**), 1 mm in (**e**–**l**).

**Figure 3 genes-10-00952-f003:**
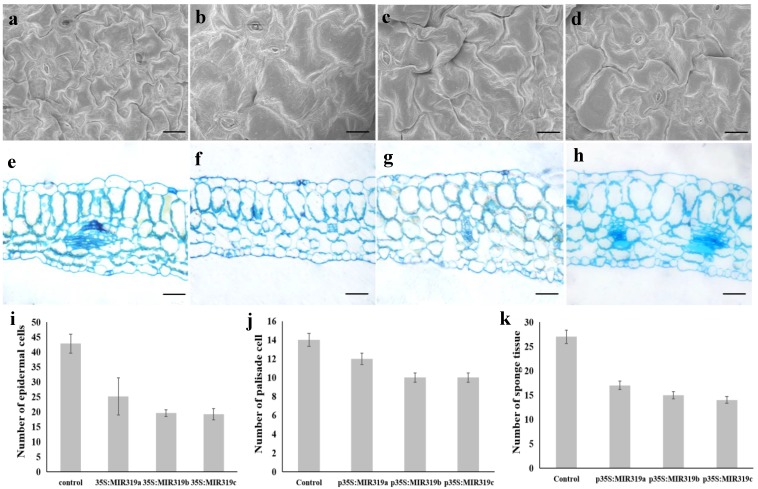
SEM and semithin section observation of leaf cells in transgenic *Arabidopsis* overexpressing the *Bra-MIR319* gene family. (**a**–**d**) SEM observation of leaf epidermal cell of wild-type plants and *Bra-MIR319* family transgenic *Arabidopsis*, respectively. (**e**–**h**) Observation of leaf structure of wild-type plants and *Bra-MIR319* family transgenic *Arabidopsis*, respectively, via semithin section analysis. (**i**–**k**) Statistical analysis of leaf epidermal cells, palisade tissue, and sponge tissue cells under 1 K magnification. Scale bars, 50 μm.

**Figure 4 genes-10-00952-f004:**
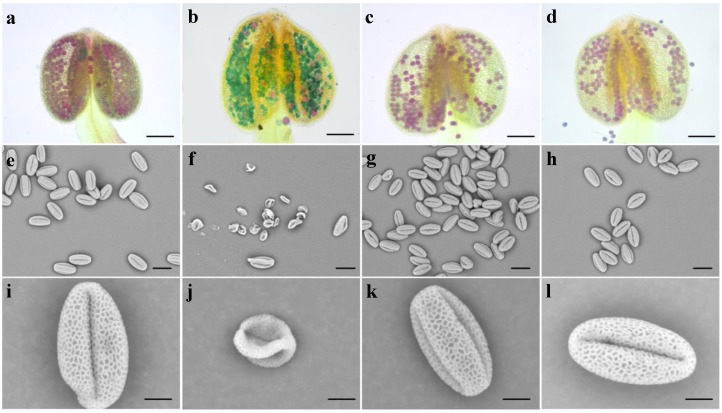
Morphological observation of pollen grains of *Bra-MIR319* family overexpressing transgenic *Arabidopsis*. (**a**–**d**) Alexander staining of anthers of wild-type and *Bra-MIR319* family transgenic *Arabidopsis*, respectively. (**e**,**i**) SEM observation of mature pollen grains of wild-type plants, (**f**,**j**) *Bra-MIR319a* transgenic plants, (**g**,**k**) *Bra-MIR319b* transgenic plants, and (**h**,**l**) *Bra-MIR319c* transgenic plants. Scale bars, 25 μm in (**a**–**h**), 50 μm in (**i**–**l**).

**Figure 5 genes-10-00952-f005:**
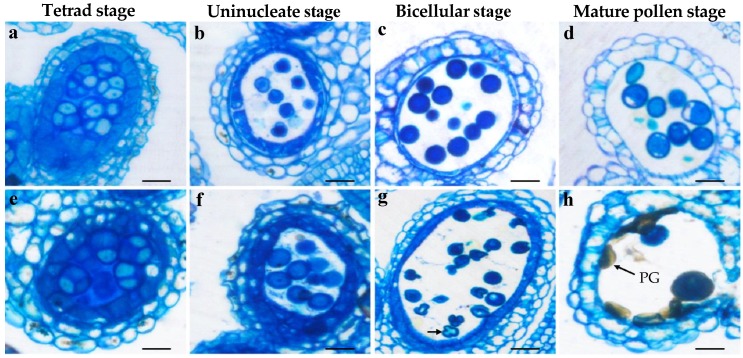
Semithin section observation of anther development in *Bra-MIR319a* overexpressing transgenic *Arabidopsis*. (**a**–**d**) Different anther development stages of wild-type plants and *Bra-MIR319a* transgenic plants (**e**–**h**). Arrows show the abnormal microspores. PG, pollen grain. Scale bars, 50 μm.

**Figure 6 genes-10-00952-f006:**
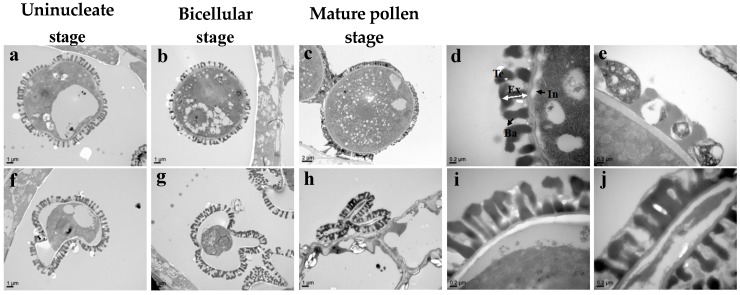
TEM observation of pollen grain in *Bra-MIR319a* overexpressing transgenic *Arabidopsis*. (**a**–**c**) Different development stages of wild-type plants and *Bra-MIR319a* transgenic plant pollen grains (**f**–**h**). (**d**,**e**) and (**i**,**j**), magnified images of (**b**,**c**) and (**g**,**h**). Te, tectum; Ba, baculum; Ex, exine; In, intine.

**Figure 7 genes-10-00952-f007:**
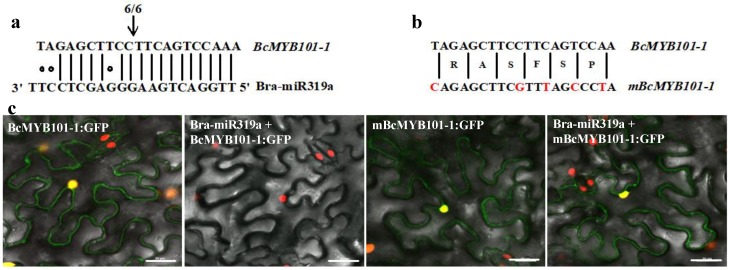
Specific regulation of *BcMYB101-1* by Bra-miR319. (**a**) Validation of the target gene *BcMYB101-1* of Bra-miR319 in *B. campestris* by 5′ RACE. (**b**) Construct of a version of BcMYB101-1: GFP with multiple mutations in the miRNA complementary motif. (**c**) Effects of Bra-miR319a on BcMYB101-1: GFP fluorescence in *Nicotiana benthamiana* leaf epidermal cells.

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
