# Peer review of "Functional Similarity and Difference among Bra-MIR319 Family in Plant Development"

_genes, 2019, doi:10.3390/genes10120952_

Round 1
Reviewer 1 Report
The manuscript entitiled "Functional redundancy and differentiation among Bra-MIR319 family in plant development" reported that miR319 from Brassica campestris play functional redundancy and differentiation in plant development including leaf cell and pollen growth. The patthern and function of miR319 found here is interesting and specific in Bracssica due to genome triplication. These results provide more knowledge for miRNA. However, all the images are required to be improved with higher resolution.
Here are some other concerns as list:
consider the text in whole manuscript "functional redundancy and differentiation" revised as "functional redundancy and difference"; Line 39: correct "different"; Line 40: check the format of "miR319a", “miR319b”, “miR319c"; Line 68: check "nuclear positioning"; Line 73: check the format of "BcUBC10", "BcU6"; Line 95, 107-109: check the format of "Bra-miR319"; Line 103: check the text "48 h"; Line 117-118: consider the sentence "Expression of Bra-MIR319a......(Figure S1)." moved to the end of line 113; Line 122: delete "grown"; check the text in Figure 1b and legend in Figure 1, "Lnf" or "Inf"? check the legend in Figure 2: what about the scal bars for i-l? Line 253: what's the meaning for "BcMYB101-1"?
Author Response
I have correct "functional redundancy and differentiation" as "Functional similarity and difference" ;
I have corrected "different" as "difference";
The format of "miR319a", "miR319b", "miR319c" mean the mature miRNA, so I don’t change the format;
I have corrected "nuclear positioning" as "nuclear localization";
I have corrected the format of "BcUBC10", "BcU6";
"Bra-miR319" mean the mature miRNA, so I don’t change the format;
I have corrected "48 h" as "48 hour";
I have moved the sentence "Expression of Bra-MIR319a......(Figure S1)." to the end of line 113;
I have corrected "Lnf" to "Inf";
The scal bars for i-l is the same as e-h, I have corrected;
BcMYB101 have three copies, BcMYB101-1 is one copy of homologs of MYB101 in B. campestris, and I add the explain in Line 201.
Reviewer 2 Report
The authors invested the functional redundancy and expression patterns of miR319 family members in Brassica campestris. The authors found Bra-MIR319 family members have functional redundancy in leaf and petal phenotypes even they have different expression patterns in Brassica organs. One of the Bra-MIR319 family members, Bra-MIR319a, showed vacuolar pollen and intine phenotypes when overexpressed in Arabidopsis.
Major comments:
One of the main conclusions the authors made is the functional redundancy of Bra-MIR319 family members. By only showing the three mutants have similar leaf and petal phenotype may not be sufficient to claim functional redundancy. To conclude functional redundancy, the authors need double/triple mutant or to show the overexpression of one MIR319 family member can rescue the knockout phenotype of another MIR319 family member. The authors may consider revise the statement about functional redundancy.
Figure 3 the resolution of these images are very low but it looks like the cell size are larger in the mutants. Would be good to measure cell size and discuss if the Bra-MIR319 have any functions in cell elongation/expansion.
Figure 3b The red nucleotides in the miRNA complementary motif are mutated to make a negative control. How these nucleotides were selected? Are the mutations at those nucleotides suppose to have a strong effect on disrupting miRNA recognition. Please describe in the method section.
Minor comments:
Labels on Figure 3 are hard to read.
Figure1a error bars represent SD or SE or 95% confidence intervals?
Figure 1b The labels on top of figures do not match the lanes.
Line 54, please cite the paper where the BLAST analysis was performed.
Line56, ‘reached over 80% ’, was over 80%
Line 60 ‘those’ instead of ‘that’
Line 86 ‘abovementioned ’ typo
Line 236 ‘could more function in petal morphogenesis. ’ grammar is not correct
Line 239 Is the dark green color in the petals of mutants cause by chlorophyll accumulation? Please make discussion.
